# Prognostic Index for Predicting Prostate Cancer Survival in a Randomized Screening Trial: Development and Validation

**DOI:** 10.3390/cancers13030435

**Published:** 2021-01-24

**Authors:** Subas Neupane, Jaakko Nevalainen, Jani Raitanen, Kirsi Talala, Paula Kujala, Kimmo Taari, Teuvo L. J. Tammela, Ewout W. Steyerberg, Anssi Auvinen

**Affiliations:** 1Unit of Health Sciences, Faculty of Social Sciences, Tampere University, FI-33014 Tampere, Finland; jaakko.nevalainen@tuni.fi (J.N.); jani.raitanen@tuni.fi (J.R.); anssi.auvinen@tuni.fi (A.A.); 2UKK Institute for Health Promotion Research, FI-33014 Tampere, Finland; 3Finnish Cancer Registry, FI-00130 Helsinki, Finland; kirsi.talala@cancer.fi; 4Department of Pathology, FIMLAB laboratory services, FI-33014 Tampere, Finland; paula.kujala@fimlab.fi; 5Department of Urology, Helsinki University Hospital, University of Helsinki, FI-00014 Helsinki, Finland; kimmo.taari@helsinki.fi; 6Department of Urology, Tampere University Hospital, University of Tampere, FI-33521 Tampere, Finland; teuvo.tammela@tuni.fi; 7Department of Public Health, Erasmus MC-University Medical Center Rotterdam, 3015 GD Rotterdam, The Netherlands; e.steyerberg@erasmusmc.nl; 8Department of Biomedical Data Sciences, Leiden University Medical Center, 2333 ZC Leiden, The Netherlands

**Keywords:** prognostic index, prediction model, prostate cancer, mortality, screening trial

## Abstract

**Simple Summary:**

A prognostic index for predicting survival of localized prostate cancer (PCa) up to 15 and 20 years was developed. The prognostic index performed well for predicting PCa survival among screened and non-screened men. The performance of the prediction model was superior to the European Association of Urology (EAU) risk groups as well as a modified cancer of prostate risk assessment (CAPRA) risk score. We further constructed a simplified risk score in an unscreened population, using the three most relevant predictors. The simplified risk score was applied to predict PCa survival at 10 years from diagnosis to provide more accurate risk estimation as the basis for decision making.

**Abstract:**

We developed and validated a prognostic index to predict survival from prostate cancer (PCa) based on the Finnish randomized screening trial (FinRSPC). Men diagnosed with localized PCa (*N* = 7042) were included. European Association of Urology risk groups were defined. The follow-up was divided into three periods (0–3, 3–9 and 9–20 years) for development and two corresponding validation periods (3–6 and 9–15 years). A multivariable complementary log–log regression model was used to calculate the full prognostic index. Predicted cause-specific survival at 10 years from diagnosis was calculated for the control arm using a simplified risk score at diagnosis. The full prognostic index discriminates well men with PCa with different survival. The area under the curve (AUC) was 0.83 for both the 3–6 year and 9–15 year validation periods. In the simplified risk score, patients with a low risk score at diagnosis had the most favorable survival, while the outcome was poorest for the patients with high risk scores. The prognostic index was able to distinguish well between men with higher and lower survival, and the simplified risk score can be used as a basis for decision making.

## 1. Introduction

Prostate cancer (PCa) presents a wide spectrum of behavior, from indolent to highly aggressive [1]. Treatment decisions are required at several phases during the course of the disease [2]. Optimal disease management should avoid both excessively aggressive treatment in patients who are not at high risk of disease progression and ineffective management of aggressive disease leading to treatment failure and development of metastatic disease. However, the dilemma expressed by Dr. Willet Whitmore persists for PCa: “Is cure possible in those for whom it is necessary—and is cure necessary in those for whom it is possible”.

Several prediction methods for the prognosis of localized PCa have been presented as tabulations [3,4,5], nomograms [6,7,8], risk groups [9,10] and decision trees [11,12]. However, these methods have mainly divided patients into 3–4 broad risk groups and used biochemical recurrence (BCR) as the end-point rather than PCa death [3,4,9,13,14]; furthermore, few are based on a modern setting with largely prostate-specific antigen (PSA)-detected cases. Prognostic prediction models based on a limited number of relevant clinical characteristics can offer evidence-based input to inform medical practice [15].

We developed and validated a full prognostic index for predicting survival of localized PCa up to 15 and 20 years. We also developed a simplified risk score tool for use at diagnosis and applied it to predict survival at 10 years. 

## 2. Results

Prognostic factors associated with PCa death included age at diagnosis, trial arm, PSA at diagnosis, European Association of Urology (EAU) risk group, treatment modality, mode of detection and biochemical recurrence (Table 1). All prognostic factors except comorbidity index showed a statistically significant difference between men who died from PCa and others.

Older age at diagnosis was marginally associated with lower PCa mortality at 9–20 years (Appendix A). PSA at diagnosis was associated with higher PCa mortality in the first and the last development periods. PCa mortality was higher in the intermediate- to high-risk groups compared to the low-risk group in all three follow-up periods. Men treated with radical prostatectomy had the most favorable survival, with the exceptions of radiotherapy and observation in the early follow-up. Biochemical recurrence also predicted increased probability of PCa death, with the largest effect after the first three years. 

The distributions of the prognostic index differed markedly across the development periods (Figure 1a–c). The graphs illustrate lower prognostic index (PI) values (indicating worse survival) for men who died from PCa than those who did not die from PCa (cumulative frequency for the former group shown as the dotted blue line above the latter group, shown as the solid red line). PCa mortality increased with increasing values of prognostic index in the initial follow-up, but after 9 years, a clear excess mortality was limited to the two highest quintiles. 

The prognostic indices were associated with PCa mortality in all EAU risk groups, though the difference was not obvious in the initial three-year period with low mortality (Appendix A). The prognostic index provided incremental information, especially in the intermediate- and high-risk groups, and its contribution was accentuated with follow-up. Furthermore, the prognostic index also predicted survival within the low-risk group in the longer follow-up.

The observed mortality matched the expected one very well at all levels of the PI during each development and validation period (Table 2). In all follow-up periods, including the validation periods, the highest quintiles of the PI showed the highest observed and expected PCa mortalities. The differences between the lower quintiles were, however, relatively small.

Less than one third of the patients remained in the initial quintile from the 0–3-year development period to the 3–6-year validation period; particularly, progression from Q1 to Q2 and Q4 to Q5 was common (Appendix A). The most frequent transition was by one step up, likely due to biochemical recurrence. However, downward transitions also occurred, reflecting changes in the regression coefficients of the variables used in the model.

The predictive ability of the prognostic indices (Figure 2) did not substantially differ between the development and validation periods: area under the curve (AUC) 0.84 (95% confidence interval, CI 0.77–0.90) for the initial development period (0–3 years) and 0.83 (0.79–0.88) for the corresponding validation period (3–6 years). Similarly, for the second development period, the AUC was 0.84 (0.81–0.88), and it was 0.83 (0.79–0.88) for the subsequent validation period. For the 9–20-year development period, the AUC was 0.83 (0.79–0.86). 

A simplified risk score at diagnosis was calculated among patients in the control arm based on the regression coefficients of three categorical parameters (age at diagnosis, PSA at diagnosis and EAU risk group) to allow easy clinical application. The simplified risk score uses a granular scale of 0 to 100, with higher score indicating increasing risk. The predictive ability of the simplified risk score at diagnosis was 0.68 (0.63–0.73) (Appendix A).

The full prognostic model displayed superior discrimination (*p <* 0.001) compared to the EAU risk group alone in all three periods (AUC for EAU risk group: 0.61, 0.53 and 0.39 during the follow-up periods of 0–3, 3–9 and 9–20 years, respectively). The simplified risk score showed superior discrimination only in the 9–20-year period (*p* < 0.001). 

The simplified risk score at diagnosis (Table 3) was used to calculate the predicted PCa survival at 10 years (Figure 3). Overall, men with a high-risk score at diagnosis had poorer survival. 

We calculated the risk score at diagnosis and 10-year survival probability among patients in both study arms, as well as performing a complete case analysis in the control arm only as a sensitivity analysis (Appendix A and Appendix A), with no substantial difference in the results.

The decision curve analysis for simplified risk score at diagnosis is presented in Figure 4. The graph gives the expected net benefit per patient relative to no PCa mortality in any patient (Treat None). The risk prediction model is of benefit for a reasonable range of 3–25%: the curve diverges only at the threshold probability of about 3%. However, the net benefit of the model is about the same as the net benefit of Treat All below 3%.

## 3. Discussion

The full prognostic index with seven variables predicted PCa mortality with a performance superior to that of the EAU risk group (AUC 0.83–0.84 vs. 0.61). The robustness of the results was confirmed by sensitivity analyses including both trial arms and omitting patients with missing data.

Our model correctly predicted the 3-year survival of 99% for the patients in the lowest quintile and 97% for those in the highest quintile of the prognostic index. We divided the follow-up time into several segments due to lack of proportionality across the entire follow-up. In the second development period, 6-year survival was 99% among men in the lowest quintile, while it was 89% among men in the highest quintile. 

Primary treatment predicted PCa mortality already in the early follow-up. The effect of biochemical recurrence increased with follow-up. Other factors did not show clear changes over the follow-up. Similar to earlier findings [16,17], we found no strong impact of comorbidity at baseline on PCa-specific survival. No earlier PCa survival prediction models have utilized the context of a randomized screening trial. Our approach enhances the applicability of the prognostic index to the current setting with widespread PSA testing. 

A simplified risk score at diagnosis was developed using three predictors selected based on their importance and interpretability in the prognostic index model. The simplified risk score is based on a granular scale ranging from 0 to 100 with three categorical variables and can be adopted in daily clinical practice with minimal data entry. 

PSA was used as a component of our prognostic index, despite being a part of the EAU risk group, because the analysis revealed that its impact was not fully captured in the EAU classification. 

Our findings are mainly in line with earlier prediction models, although patient populations, outcomes and methodological approaches differ between studies. The performance of our simplified risk score tool at diagnosis was superior to that of the D’Amico risk classification and EAU risk group (AUC 0.68 vs. 0.59 and 0.61, respectively). The simplified risk score also outperformed an abridged version of the cancer of prostate risk assessment (CAPRA) risk score (AUC 0.59), though we were unable to incorporate percentage cancer in biopsy for estimating the CAPRA score in our analysis due to lack of data. Furthermore, all patients in our study were aged > 50 years at diagnosis [7]. 

An earlier study presented a clinical–genomic risk group classification for localized prostate cancer that showed a 10-year rate for distant metastases of 3.5% for a low-risk group, while it was 58% in a high-risk group in the training cohort, and the corresponding values for the validation cohort were 0% and 63%, respectively [9]. That risk group required extensive genomic data, restricting its applicability. Peters et al. [18] developed a prediction model for recurrent disease with three categories, which showed 60% biochemical disease-free survival and 40% composite end-point-free survival at 4 years for a low-risk group, while the corresponding figures for a high-risk group were 7% and 0%, respectively.

Decision curve analysis shows the benefit of use of the prediction model (simplified risk score at diagnosis). A net benefit was found for a reasonable range of 3–25%: the curve diverged only at the threshold probability of about 3%. 

Our study had also some limitations. The patients were treated during a period spanning from the 1990s into the 2010s and treatment modalities have evolved over time. However, long follow-up is required due to the favorable prognosis to capture the full natural course of the disease and accrue a sufficient number of PCa deaths. Completeness of data was high, with the highest proportion of missing data for PSA, at 3%. However, we used imputation in the main analysis, and sensitivity analyses of complete cases yielded comparable results, suggesting that this did not affect our findings. We incorporated biochemical relapse in the prognostic index, even though in the clinical setting, it is not available at diagnosis. On the other hand, its inclusion enhances the applicability of our results in prognostic prediction after the initial phase and post-primary treatment.

## 4. Material and Methods

We used data from the Finnish Randomized Study of Screening for Prostate Cancer (FinRSPC). The trial protocol and main results have been described elsewhere [19]. In brief, a random sample of 8000 men aged 55–67 years were allocated to the screening arm (SA) annually in 1996–1999 and the remaining men (48,278 in total) formed the control arm that received no intervention. Men in the screening arm were invited for screening based on serum PSA. Screen-positive men (defined as those with PSA ≥ 4.0 ng/mL or PSA 3.0–3.9 ng/mL with free–total PSA ratio < 0.16) were referred to a local urological clinic for diagnostic examinations including transrectal ultrasound-guided biopsy. The second screening round was conducted four years later, and the final one after 8 years. Men aged > 71 years, those diagnosed with prostate cancer and men who had emigrated from the study area were no longer invited. 

All men diagnosed with localized prostate cancer between randomization and the end of 2015 were included in this analysis (N = 7042). The follow-up for the primary analysis started at diagnosis and ended at death, emigration or the common closing (31 December 2015). Death from prostate cancer was the end-point in the analysis, with underlying causes of death obtained from Statistics Finland.

Information on tumor, lymph node, and metastasis (TNM) stage and Gleason score were abstracted from medical records. For previous cases, Gleason scores were revised according to the 2002 system by two pathologists. PSA at diagnosis was used for all men. Information on biochemical recurrence was obtained from laboratory databases. Biochemical recurrence (BCR) was defined as PSA reaching at least 0.2 ng/mL in two measurements after prostatectomy, while BCR after radiotherapy was defined as a rise in PSA by at least 2.0 ng/mL above the lowest level (nadir). A modified version of the Charlson comorbidity index [20] was constructed based on hospital inpatient episodes obtained from the nationwide hospital discharge registry and categorized into no versus any comorbidity (score 0 versus 1 to 8) [21]. Prognostic risk group for PCa survival at diagnosis was classified as low, moderate and high, according to the European Association of Urology (EAU) criteria [10]. Low-risk PCa was defined as stage T1–T2a with Gleason score < 7 and PSA < 10 ng/mL; intermediate risk as T1–T2b with either Gleason 7 or PSA 10–20; high risk was stage T1–T2c with either Gleason > 7 or PSA 20–100, or T2c. 

Primary treatment was retrieved from medical records and classified as radical prostatectomy, curative radiation therapy (external beam or brachytherapy), endocrine therapy (luteinizing hormone-releasing hormone agonist/antagonist, anti-androgen, or both or surgical castration), observation (watchful waiting or active surveillance) or no treatment. 

### 4.1. Ethical Issues

Helsinki and Tampere University Hospital Ethics committees reviewed the study protocol (tracking number R10167). Cancer registry data were obtained with permission from the National Institute for Health and Welfare (Dnro THL/1601/5.05.00/2015). Written informed consent was obtained from the men participating in the screening arm.

### 4.2. Statistical Analysis

For the preliminary investigation, the proportional hazards assumption of the factors in the Cox regression was evaluated by graphical examination in log–log plots. These plots formed approximate parallel straight lines as required, except those for primary treatment which crossed each other. For this reason, we divided the follow-up time into three periods (0–3, 3–9 and 9–20 years) to model the effect of the full prognostic index separately during each period. 

We used a same set of variables (age at diagnosis, study arm, PSA at diagnosis, EAU risk group, comorbidity index, primary treatment and biochemical recurrence) in a complementary log–log regression model, identified by a stepwise forward selection with *p* = 0.10 as the cut-off, in each of the three periods. Only statistically significant interaction terms (5% level) were included in the final model. The prognostic index (PI) was then derived as a linear combination of the variables, including interaction terms and their coefficients from the regression model. We generated prognostic indices separately for the three follow-up periods, hereafter called development periods, using regression coefficients estimated from the complementary log–log models. The probabilities of PCa death during each development period using prognostic indices were calculated. 

Missing values (3.1% in PSA and 2.6% in the EAU risk group) were imputed using a multiple imputation by chained equations (MICE) algorithm, assigning multiple likely values from a predicted distribution based on association with other variables [22]. Multiple imputation creates multiple copies of the dataset, which are analyzed separately. Finally, the results were appropriately combined [23].

To avoid overfitting and overestimation of the predictive ability, we validated the results by applying them to a subsequent follow-up period for the first two development periods (i.e., prognostic index derived from the development period of 0–3 years from diagnosis to predict survival during the validation period of 3–6 years, and the index derived from the development period of 3–9 years to predict survival during years 9–15) [24]. Expected and observed probabilities and numbers of PCa deaths were calculated for each development and validation period. Expected probability of PCa death for the validation period was calculated as the inverse of the complementary log–log transformation.

Reclassification probabilities of men in the quintiles of prognostic index for the first development period (0–3 years) and validation period (3–6 years) were calculated. Moreover, we presented the distribution and the mean values of full prognostic indices by risk group for all development periods. Cumulative distribution of prognostic index values according to the survival status of the patients for all three development periods was plotted. 

Receiver operating characteristic (ROC) was calculated for the test and validation periods to illustrate sensitivity and specificity. The area under the curve (AUC) was calculated to assess the discriminative power of the prediction models. We further developed a simplified risk score at diagnosis using the information at diagnosis (age at diagnosis, PSA at diagnosis and EAU risk group) for the control arm only to avoid lead-time by screening. Only three variables were selected based on their importance and interpretability in the prognostic index model. The simplified risk score was then used to calculate the predicted probability of 10-year PCa survival and was presented graphically. We further developed the decision curve analysis to determine the clinical usefulness of a simplified risk score at diagnosis by quantifying the net benefits at different threshold probabilities.

As a sensitivity analysis, we calculated the risk score at diagnosis based on data in both study arms. Furthermore, we performed a complete case analysis in the control arm by calculating the risk score and predicted 10-year PCa survival to examine the potential influence of imputation.

Analyses were performed using Stata Statistical Software version 16.0 (StataCorp, College Station, TX, USA) and IBM SPSS Statistics 23 (IBM Corp., Armonk, NY, USA). 

## 5. Conclusion

The prognostic index accurately predicted prostate cancer survival at follow-up reaching 20 years. A simplified risk score at diagnosis using the three most relevant parameters to predict the survival at 10 years can be helpful for providing more accurate risk estimation as the basis for decision making. However, our prediction model requires further external validation.

## Figures and Tables

**Figure 1 cancers-13-00435-f001:**
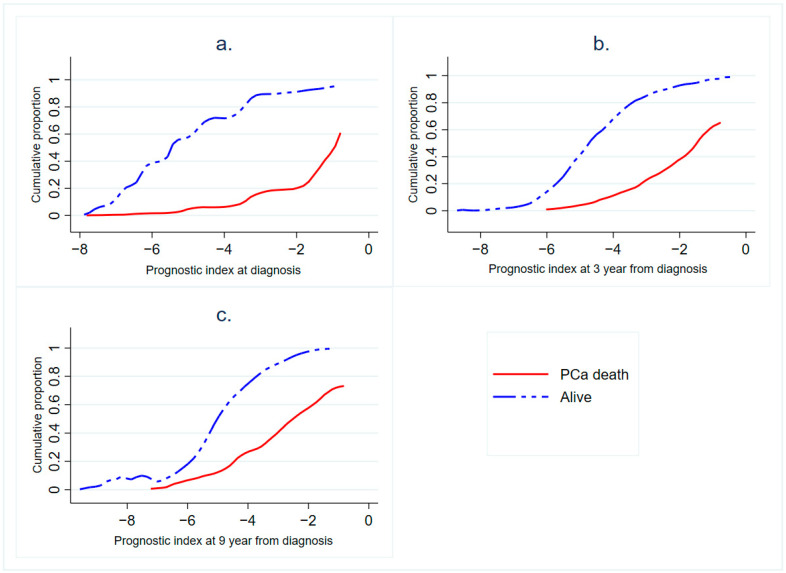
Cumulative distribution of prognostic index (PI) values for men who died due to prostate cancer (PCa) and those who did not during (**a**) the development period of 0–3 year (**b**) the development period of 3–9 years and (**c**) the development period of 9–20 years. The distribution of men is described as a cumulative frequency across values of the PI from low (indicating worse survival) to high (indicating favorable survival).

**Figure 2 cancers-13-00435-f002:**
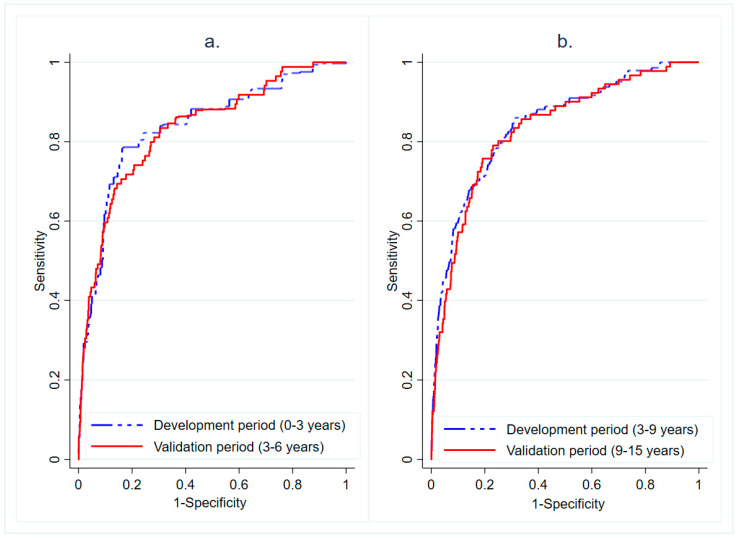
(**a**) Receiver operating characteristic (ROC) curves for the development period of 0–3 year and validation period of 3–6 years and the (**b**) development period of 3–9 years and validation period of 9–15 years.

**Figure 3 cancers-13-00435-f003:**
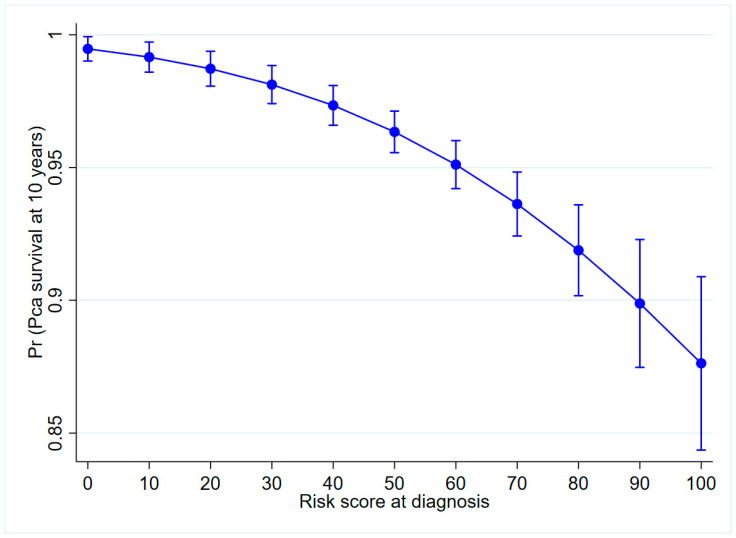
Predicted probability of PCa survival at 10 years from diagnosis among controls in the Finnish randomized screening trial.

**Figure 4 cancers-13-00435-f004:**
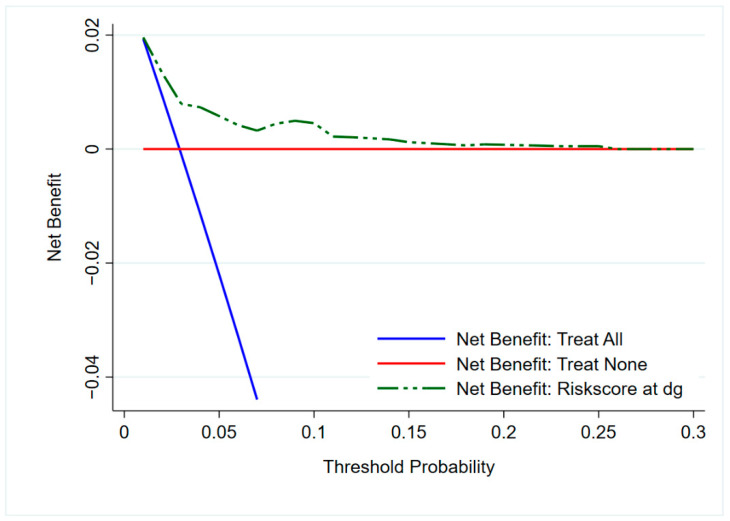
Decision curve analysis for the simplified risk score at diagnosis. The dotted green line is the simplified risk score (prediction model), the blue solid line assumes PCa mortality in all patients and red solid line assume no patient deaths due to PCa. Threshold probability on the *x*-axis is the level of diagnostic certainty above which the patient would choose to be treated.

**Table 1 cancers-13-00435-t001:** Demographic and clinical characteristics of 7042 localized prostate cancer (PCa) patients in the cohort stratified by survival status at 20-year follow-up from the date of diagnosis.

Characteristic	Total (*N* = 7042)	No PCa Death ^†^ (*n* = 6737)	PCa Death (*n* = 305)	*p*-Value ^‡^
**Age at entry (years)**				<0.001
55	1844	1806 (97.9%)	38 (2.1%)	
59	1898	1840 (96.9%)	58 (3.1%)	
63	1782	1681 (94.3%)	101(5.7%)	
67	1518	1410 (92.9%)	108 (7.1%)	
**Age at diagnosis (years)**				<0.001
Median (IQR)	69 (65–73)	69 (65–73)	68 (64–71)	
**Study arm**				
Control	3823	3667 (95.9%)	156 (4.1%)	
Screening	3219	3070 (95.4%)	149 (4.6%)	
**PSA at diagnosis (ng/mL)**				<0.001
Median (IQR)	7.8 (5.2–11.7)	7.7 (5.2–11.5)	10.3 (6.4–18.6)	
**Biopsy Gleason sum**				<0.001
2–6	4031	3913 (97.1%)	118 (2.9%)	
7	2249	2148 (95.5%)	101 (4.5%)	
8–10	705	630 (89.4%)	75 (10.6%)	
Missing	57	46 (80.7%)	11 (19.3%)	
**EAU risk group**				<0.001
Low	2769	2712 (97.9%)	57 (2.1%)	
Intermediate	2988	2864 (95.9%)	124 (4.2%)	
High	1285	1161 (90.4%)	124 (9.7%)	
Missing	236	225 (95.3%)	11 (4.7%)	
**Comorbidity index**				0.309
0	6320	6041 (95.6%)	279 (4.4%)	
1+	722	696 (96.4%)	26 (3.6%)	
**Primary treatment**				<0.001
Radical Prostatectomy	1812	1751 (96.6%)	61 (3.4%)	
Radiation	2718	2583 (95.0%)	135 (5.0%)	
Endocrine	643	570 (88.7%)	73 (11.4%)	
Observation	1788	1756 (98.2%)	32 (1.8%)	
No treatment	78	74 (94.9.6%)	4 (5.1%)	
Missing	3	3 (100.0%)	0	
**Method of presentation**				0.011
Screen-detected	1462	1381 (94.5%)	181 (5.5%)	
Not screen-detected	5578	5354 (96.0%)	224 (4.0%)	
Missing	2	2 (100.0%)	0	
**Biochemical recurrence**				<0.001
No	4749	4672 (98.4%)	77 (1.6%)	
Yes	2212	1988 (89.9%)	224 (10.1%)	
Missing	81	77 (95.1%)	4 (4.9%)	

IQR: Interquartile range. ^†^ Includes men alive and deaths due to causes other than PCa. ^‡^
*p*-values for categorical variables were derived from a chi-square test, whereas for continuous variable, using ANOVA test.

**Table 2 cancers-13-00435-t002:** Expected and observed probability and number of PCa deaths in the development and validation periods for quintiles of prognostic index.

Prognostic Index Quintiles	Men	Development Period						Validation Period			
		Number of Deaths ^†^	Observed		Expected		Men	Observed		Expected	
			Probability ^‡^	PCa Deaths	Probability ^‡^	PCa Deaths		Probability ^‡^	PCa Deaths	Probability ^‡^	PCa Deaths
	0–3 year						3–6 years				
Q1	1436	67	0.001	1	0.001	1	1323	0.001	1	0.001	1
Q2	1446	87	0.003	4	0.002	3	1352	0.005	7	0.003	4
Q3	1438	44	0.003	4	0.003	4	1354	0.003	4	0.004	5
Q4	1436	66	0.002	3	0.006	9	1299	0.010	13	0.008	10
Q5	1465	175	0.031	46	0.028	41	1396	0.044	61	0.044	62
Total	7221	439	0.008	58	0.008	58	6724	0.013	86	0.012	82
	3–9 years							9–15 years			
Q1	1319	118	0.002	3	0.002	3	1133	0.002	2	0.001	1
Q2	1353	165	0.007	9	0.004	6	1155	0.005	6	0.003	4
Q3	1356	150	0.005	7	0.007	10	1147	0.004	4	0.004	5
Q4	1319	176	0.018	24	0.014	19	1152	0.009	10	0.011	13
Q5	1377	95	0.074	102	0.075	103	1248	0.054	69	0.047	61
Total	6724	704	0.022	145	0.021	141	5875	0.016	91	0.014	84
	9–20 years										
Q1	1122	26	0.002	2	0.002	2					
Q2	1148	55	0.004	5	0.004	5					
Q3	1133	93	0.007	8	0.006	7					
Q4	1180	132	0.019	22	0.014	17					
Q5	1292	126	0.057	74	0.054	70					
Total	5875	432	0.019	111	0.017	101					

^†^ Number of deaths due to causes other than PCa; ^‡^ probability of PCa death for an individual man from the prognostic model.

**Table 3 cancers-13-00435-t003:** Scoring rules for constructing the simplified risk score at diagnosis for PCa survival based on the complementary log–log regression model among PCa-diagnosed cases in the control arm.

Characteristics	Categories	β ^†^	Risk Score ^§^
Age at diagnosis (years)			
	≤60	1.12	40
	61–70	1.00	35
	71–75	0.72	25
	≥76	Ref	0
PSA at diagnosis (ng/mL)			
	≤19.9	Ref	0
	≥20.0	0.22	10
EAU risk group			
	Low	Ref	0
	Intermediate	0.75	30
	High	1.46	50

^†^ Regression coefficients from complementary log–log model. ^§^ Risk score at baseline calculated by dividing each beta coefficient by the sum of the highest beta coefficient of each variable and then multiplied by 100 (scores are rounded).

## Data Availability

The data presented in this study are available on request from the corresponding author.

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
