# Peer review of "Prognostic Index for Predicting Prostate Cancer Survival in a Randomized Screening Trial: Development and Validation"

_cancers, 2021, doi:10.3390/cancers13030435_

Round 1
Reviewer 1 Report
The article entitled “Prognostic index for predicting prostate cancer survival in a randomized screening trial: Development and validation” by Neupane et al. aimed to develop a risk model for prostate cancer death prediction.
The article would be of interest for onco-urologists; however, there are some points that need to be addressed before publication.
- Both proposed models include the EUA risk groups and PSA at diagnosis. Although the authors discuss that “the analysis revealed that its [PSA] impact was not fully captured in the EAU classification” and therefore the included PSA at diagnosis as an additional variable in their predictive model, I believe this is not correct since they included the same variable twice in the model. For example, in the simplified model they used a PSA cut-off of 20 ng/mL, which is the same cut-off used in the EUA model to categorized patients as high risk. This might be the reason why the score for PSA>20ng/mL is only 5/100 when all other variables scored >25/100 points. I suggest removing PSA from the models, unless they demonstrate that the inclusion of PSA significantly improve the prediction.
- In line with the comment above, when the authors describe the ROC analysis for the simplified model (lines 98-100) they state that their model displayed superior discrimination compared to EUA. Please provide the p-value for the comparison of AUCs to support this conclusion.
- For figures 3 and S3, they combined patients with hormonal treatment or no treatment. What was the rationale for combining these two groups? Did they expected/observed that patients with hormonal treatment had the same 10-year survival than patients receiving no treatment? It is not clear why they did this. Please explain it in the text.
- The authors seemed to use “Age” and “PSA” as continuous variables in the full model; but they used them as categorical variables in the simplified model. Please explain why there was a change in the definition of the variables between both models and how the cut-offs were chosen.
- Table S2 is hard to read. In addition, in lines 79 through 81, the authors state that they observed a progression from lower to higher quantiles in the prognostic index 0-3 years and 3-6 years probably due to biochemical relapse. However, it seems that there was also a downstaging for some patients. For example, 589 (42.7%) patients were classified as Q2 by the prognostic index 0-3 years; but they were classified as Q1 by the prognostic index 3-6 years. The same happens for all other quantiles. Please describe and discuss this downstaging.
Minor comments:
- In table 1 it is not clear what the categories for “Age at entry” are. Are they ranges from the age stated in the table to age of the next category (e.g. 55-58, 59-62, etc)? Please correct this in the table to make it clearer.
- Please add the p-values for differences between “No PCa death” and “PCa death” groups in table 1.
Author Response
# Reviewer 1:
The article would be of interest for onco-urologists; however, there are some points that need to be addressed before publication.
1. Both proposed models include the EUA risk groups and PSA at diagnosis. Although the authors discuss that “the analysis revealed that its [PSA] impact was not fully captured in the EAU classification” and therefore the included PSA at diagnosis as an additional variable in their predictive model, I believe this is not correct since they included the same variable twice in the model. For example, in the simplified model they used a PSA cut-off of 20 ng/mL, which is the same cut-off used in the EUA model to categorized patients as high risk. This might be the reason why the score for PSA>20ng/mL is only 5/100 when all other variables scored >25/100 points. I suggest removing PSA from the models, unless they demonstrate that the inclusion of PSA significantly improve the prediction.
Response: We thanks the reviewer for this comment. We used a set of variables in complementary log-log regression model, identified by a stepwise forward selection with p=0.10 as the cut-off in the full data set. All variables including PSA at diagnosis selected for the model have a statistically significant association with PCa death. As it also explained in the manuscript, PSA at diagnosis was not fully captured in the EAU classification and therefore we log transformed the PSA value at diagnosis in the full model. In the simplified, categorical PSA at diagnosis was included in the risk score to improve its applicability (assigning a value for the score based a fixed cut-off is easier than calculating a score as a multiple of a transformed PSA value). Just as a minor correction, the risk score for PSA at diagnosis <20 ng/ml is actually 10 on a scale of 0-100, not 5 out of 0-100, please see Table 3.
2. In line with the comment above, when the authors describe the ROC analysis for the simplified model (lines 98-100) they state that their model displayed superior discrimination compared to EUA. Please provide the p-value for the comparison of AUCs to support this conclusion.
Response: The difference is statistically significant (<0.001) with the full index, and the simplified risk score only at 9-20 years period (<0.001). We have now provided the p-value for the comparison in text.
3. For figures 3 and S3, they combined patients with hormonal treatment or no treatment. What was the rationale for combining these two groups? Did they expected/observed that patients with hormonal treatment had the same 10-year survival than patients receiving no treatment? It is not clear why they did this. Please explain it in the text.
Response: Initially, we in fact created the Figure 3 and Figure S3 with hormonal treatment and no treatment group separately. However, the two groups had no difference in 10-year survival (the curves actually overlapped) and therefore, the two groups were combined. We have now clarified this in the text, added text in line 115-116 “We found no major differences between men receiving hormonal treatment and no treatment, therefore two groups were combined in the analysis”.
4. The authors seemed to use “Age” and “PSA” as continuous variables in the full model; but they used them as categorical variables in the simplified model. Please explain why there was a change in the definition of the variables between both models and how the cut-offs were chosen.
Response: We thank the reviewer for pointing this out. This issue was also addressed for PSA in Comment #1 and in our response to it. Indeed, age and PSA variables were used as categorical variables in the simplified risk score model. The reason was to improve applicability of the risk score, as categorical variables are more conveniently applicable in a clinical setting (simply choosing a value from the table) than a linear function (age x coefficient). The cut-off values for both the variables were chosen based on earlier literature, as well as the distribution of the variables in the current data. We initially used three categories for PSA (<10ng/ml, 10.0-19.9ng/ml and ≥20ng/ml), but second category (10-19.9ng/ml) had the same risk score as of the reference group, therefore we decided to use two categories only.
5. Table S2 is hard to read. In addition, in lines 79 through 81, the authors state that they observed a progression from lower to higher quantiles in the prognostic index 0-3 years and 3-6 years probably due to biochemical relapse. However, it seems that there was also a downstaging for some patients. For example, 589 (42.7%) patients were classified as Q2 by the prognostic index 0-3 years; but they were classified as Q1 by the prognostic index 3-6 years. The same happens for all other quantiles. Please describe and discuss this downstaging.
Response: The distribution of prognostic index quintiles for the development period 0-3 years is reclassified in new prognostic index quintiles for the validation period 3-6 years. We found the most frequent transition is one step up, which is likely due to biochemical recurrence. However, as the reviewer correctly pointed out, there are transitions one step down probably due to changes in the regression coefficient of the variables used in the model. We have now added a sentence in the manuscript to note this, added text: “However, there was some unusual downward transition maybe due to changes in the regression coefficient of the variables used in the model”, in line 87-89.
Minor comments:
6. In table 1 it is not clear what the categories for “Age at entry” are. Are they ranges from the age stated in the table to age of the next category (e.g. 55-58, 59-62, etc)? Please correct this in the table to make it clearer.
Response: Age at entry is the age of men at the time they entered to the screening trial, which is exact age of 55, 59. 63 and 67.
7. Please add the p-values for differences between “No PCa death” and “PCa death” groups in table 1.
Response: P-value for the difference between No PCa death and PCa death has now been added in Table 1 as suggested.
Reviewer 2 Report
The author developed a Simplified Risk Score combining EAU Risk Group, Age, and PSA at diagnosis. That is rather a simple way to predict the prognosis with high predictive value with a long follow-up period.
Major
- In the case of localized cancer, the T stage is critical. The initial factors included looks rough and insufficient to develop the model. Why not included the T stage.
- The result of Figure 1 looks not easy to understand. Could you describe in more detail how to read the figures?
- Regarding the development period and validated periods, the author used the same cohort but at a different time point. The author needs to use external cohort to validate the model.
- Figure 3 compares the prognosis based on the treatment modules. This is very missleading. Since there are so many patients with different tumor stages and PSA values between groups, please provided a comparison after the propensity-matched cohort.
Author Response
Reviewer #2:
The author developed a Simplified Risk Score combining EAU Risk Group, Age, and PSA at diagnosis. That is rather a simple way to predict the prognosis with high predictive value with a long follow-up period.
Response: We thank the reviewer for this assessment.
Major
1. In the case of localized cancer, the T stage is critical. The initial factors included looks rough and insufficient to develop the model. Why not included the T stage.
Response: This seems like a misunderstanding, in fact, T stage is incorporated in the EAU risk group. Please see the text in the methods part line 207-210.
2. The result of Figure 1 looks not easy to understand. Could you describe in more detail how to read the figures?
Response: Table 1 provides the cumulative distribution of prognostic index values from the full index for men who died due to PCa vs alive. The main message is that the prognostic index is able to distinguish men who subsequently die from PC form the other group. We have now added some text in the table heading of Table 1 and in the manuscript to clarify this, added text “distribution of men is described as cumulative frequency across values of the PI from low (indicating worse survival) to high (indicating favorable survival)”. Please see the added text in the table title, page 3.
3. Regarding the development period and validated periods, the author used the same cohort but at a different time point. The author needs to use external cohort to validate the model.
Response: Thank you for this comment. We agree that ideally, a separate validation study should be conducted with independent data. It took us two decades to compile this material and we had no access to any comparable dataset. Hence, dividing the follow-up into separate periods is the best we were able to do currently, and this is also mentioned in the conclusions. Moreover, the current approach achieved a better statistical power compared with dividing the data into a development and validation sets of subjects.
4. Figure 3 compares the prognosis based on the treatment modules. This is very misleading. Since there are so many patients with different tumor stages and PSA values between groups, please provided a comparison after the propensity-matched cohort.
Response: We thank the reviewer for this concern. Ideally, of course treatment comparisons should be based on randomization. Our study was randomized screening trial, but patient selection for treatment followed normal clinical practice. Hence, this is a pragmatic analysis of the real-world data, but we are aware of the fact that the comparability of the patient groups across treatment modalities is questionable. Therefore, comparisons between treatment s are presented in a descriptive manner, simply stating the outcome, without claiming superiority or inferiority for any of the treatment options knowing that the comparisons may not be fair. An analysis using propensity-matched cohorts is an option for more refined analysis and we would be keen to pursue this suggestion, but we were not able to perform such analysis in a short period of time (10 days, including Christmas holidays) allocated for the revision and also considering the fact that this is not the main goal of the paper. If the reviewer feels the results are misleading, we are willing to drop the comparison of treatment modalities and focus on the prognostic index only.
Round 2
Reviewer 1 Report
I thank the authors for addressing all my comments and suggestions. I believe that the manuscript is suitable for publication.
Author Response
Thank you very much for your positive feedback.